# BEEM-Static: Accurate inference of ecological interactions from cross-sectional microbiome data

**Chenhao Li**[1,2]*, **Tamar V. Av-Shalom**[1,3], **Jun Wei Gerald Tan**[1], **Junmei Samantha Kwah**[1], **Kern Rei Chng**[1], **Niranjan Nagarajan**[1,4,5]*

**1** Laboratory of Metagenomic Technologies and Microbial Systems, Genome Institute of Singapore, Singapore, Singapore, **2** Center for Computational and Integrative Biology, Massachusetts General Hospital, Boston, Massachusetts, United States of America, **3** Department of Cell & Systems Biology, University of Toronto, Toronto, Canada, **4** School of Computing, National University of Singapore, Singapore, Singapore, **5** Yong Loo Lin School of Medicine, National University of Singapore, Singapore, Singapore

\* cli40@mgh.harvard.edu (CL); nagarajann@gis.a-star.edu.sg (NN)

**Data Availability Statement:** BEEM-Static is available as an R package at https://github.com/CSB5/BEEM-static under the MIT license.

**Funding:** CL, TVAS, JWGT, JSK, KRC and NN are supported by Agency for Science, Technology and

## Abstract

The structure and function of diverse microbial communities is underpinned by ecological interactions that remain uncharacterized. With rapid adoption of next-generation sequencing for studying microbiomes, data-driven inference of microbial interactions based on abundance correlations is widely used, but with the drawback that ecological interpretations may not be possible. Leveraging cross-sectional microbiome datasets for unravelling ecological structure in a scalable manner thus remains an open problem. We present an expectation-maximization algorithm (BEEM-Static) that can be applied to cross-sectional datasets to infer interaction networks based on an ecological model (generalized Lotka-Volterra). The method exhibits robustness to violations in model assumptions by using statistical filters to identify and remove corresponding samples. Benchmarking against 10 state-of-the-art correlation based methods showed that BEEM-Static can infer presence and directionality of ecological interactions even with relative abundance data (AUC-ROC>0.85), a task that other methods struggle with (AUC-ROC<0.63). In addition, BEEM-Static can tolerate a high fraction of samples (up to 40%) being not at steady state or coming from an alternate model. Applying BEEM-Static to a large public dataset of human gut microbiomes (n = 4,617) identified multiple stable equilibria that better reflect ecological enterotypes with distinct carrying capacities and interactions for key species.

## Conclusion

BEEM-Static provides new opportunities for mining ecologically interpretable interactions and systems insights from the growing corpus of microbiome data.

## Author summary

Characterizing the ecological interactions among microbial members is an important step towards understanding the structure and function of diverse microbial communities.

Research (A*STAR) and an Industry Alignment Fund - Prepositioning Programs (IAF-PP) grant (H18/01/a0/016) from the Biomedical Research Council (BMRC). The funders had no role in study design, data collection and analysis, decision to publish, or preparation of the manuscript.

**Competing interests:** The authors have declared that no competing interests exist.

Widely used correlation based approaches for inferring interactions from cross-sectional microbiome sequencing data are not able to predict the directionality of interactions, and their results may not be interpretable. We developed an expectation-maximization algorithm (BEEM-Static) that can infer directed interaction networks from cross-sectional data based on an ecological model. Our benchmarking results showed that BEEM-Static inferred presence and directionality of interactions accurately, while correlation based methods had performance slightly better than random guesses. In addition, BEEM-Static was robust to various types of noises using statistical filters to identify and remove data points violating its assumptions. Applying BEEM-Static to a large public dataset of human gut microbiomes, we were able to identify multiple stable equilibria with distinct ecological properties.

## 1. Introduction

Microbial communities represent complex systems that impact various aspects related to human health, e.g. agriculture [1], food processing [2], disease biology [3, 4] and healthcare [5]. Interactions between members of a microbial community determine emergent phenomena such as homeostasis in the ecosystem [6, 7] and overall function of the microbiome [8]. Ecological interactions can be grouped into six major categories including mutualism (positive-positive), competition (negative-negative), antagonism (positive-negative, further includes predation and parasitism), commensalism (positive-neutral), amensalism (negative-neutral) and neutralism (neutral-neutral) [9]. Correspondingly, ecological modeling of microbiomes taking into account such interactions is a key step towards understanding community function [10, 11], forecasting dynamics [12, 13] and rationally designing interventions that alter community structure and function [14].

Advances in high-throughput sequencing and metagenomics have enabled several data-driven approaches to infer microbial interactions, bypassing limitations of experimental approaches in terms of time, resources and cultivability [9, 15]. In particular, correlation-based methods are widely used for their convenient applicability to cross-sectional datasets [9, 16], despite their inability to capture directionality of ecological interactions such as predation and parasitism [9]. Recent studies have also highlighted other pitfalls in correlational analysis, particularly the accuracy of interactions identified even when the data reflects known modes of microbial interactions [17, 18].

Predictive and dynamic modeling of microbiomes based on first-order differential equations (e.g. with generalized Lotka-Volterra models or gLVMs) has found increasing usage and provided useful insights into microbial interactions and dynamics [10, 19, 20]. Wider adoption of such techniques has been hampered by the need for large datasets (as the number of parameters grows quadratically with the number of species) and dense longitudinal sampling to adequately capture fine-grained dynamics [21]. Theoretical assumptions such as the availability of data where all species are at equilibrium (i.e. abundances of species will not change without external perturbation), and where absolute abundances are accurately known, make the determination of gLVM parameters from cross-sectional data solvable in principle [22]. In practice, microbiome data generated from high-throughput sequencing (16S rRNA gene amplicon or whole metagenomic sequencing) provides relative abundances and scaling these accurately enough for gLVM parameter estimation can be challenging [23]. Furthermore, real-world datasets often contain a mixture of perturbed and unperturbed microbiomes where the equilibrium status is unknown, and where data may even come from multiple models [24].

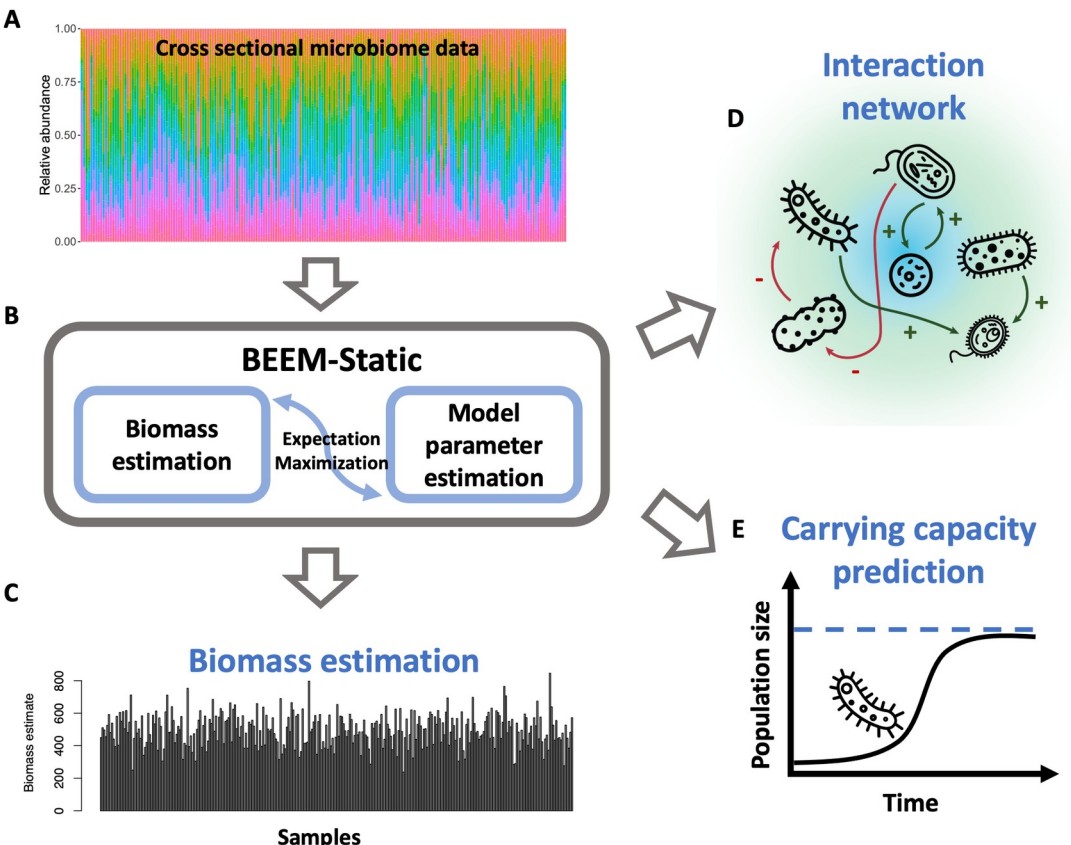

**Fig 1. Schematic overview for the BEEM-Static algorithm.** BEEM-Static takes relative abundances from a cross-sectional microbiome dataset (A), and runs an expectation-maximization algorithm (B) to estimate both biomass values (C), the interaction network (D) and carrying capacities (E).

We have previously introduced an expectation maximization (EM) algorithm which couples gLVM parameter and scaling factor estimation (BEEM), specifically for longitudinal microbiome data [23]. Here we transform this EM algorithm to work with cross-sectional data from communities that are at or near equilibrium (BEEM-Static; **Fig 1**). In benchmarking comparisons with simulated communities against 10 other methods that infer microbial interactions from cross-sectional data, we noted that while all other methods only improved slightly over random predictions (AUC-ROC<0.63), BEEM-Static exhibited high accuracy similar to estimation using true scaling values (AUC-ROC>0.88). Similar observations were made with synthetic communities based on all-pair co-culture experiments, where BEEM-Static accurately recapitulated nearly all known interactions and their directionality. Based on statistical filters to identify non-model and/or non-equilibrium samples in real and simulated datasets, we show that BEEM-Static can be robust to up to 40% of data violating these assumptions. Applying BEEM-Static to a large public collection of human gut microbiome profiles (n = 4,617) identified multiple stable equilibria that appear to better reflect ecological enterotypes with distinct carrying capacities and interactions for key species (e.g. *Prevotella copri*) compared to prior clustering based definitions [24]. BEEM-Static thus provides new opportunities for mining ecologically interpretable interactions and systems insights from the growing corpus of microbiome data in the public domain.

## 2. Methods

### 2.1 Estimating biomass and gLVM parameters with cross-sectional data

The gLVM model is a set of differential equations describing the instantaneous growth rate of each species ($dx_i(t)/dt$) as a function of absolute cell densities ($x_i(t)$) of the $p$ species in a community:

$$\frac{dx_i(t)}{dt} = \mu_i x_i(t) + \sum_{j=1}^{p} \beta_{ij} x_i(t) x_j(t)$$

where $\mu_i$ is the intrinsic growth rate of species $i$ and $\beta_{ij}$ are interaction terms that define the strength of the influence of species $j$'s abundance on species $i$'s growth. In general, estimating gLVM parameters ($\mu_i$ and $\beta_{ij}$) requires longitudinal data to measure $dx_i(t)/dt$. However, at the non-trivial equilibrium ($dx_i(t)/dt = 0$ and $x_i > 0$):

$$\mu_i + \sum_{j=1}^{p} \beta_{ij} x_j = 0,$$

where the time parameter $t$ now becomes implicit in the equation. Dividing both sides by $-\beta_{ii}$ and the biomass $m$, rearranging terms, and re-parameterizing the equation we get:

$$\tilde{x}_i = \frac{a_i}{m} + \sum_{j=1, j \neq i}^{p} b_{ij} \tilde{x}_j,$$

where $a_i = -\mu_i / \beta_{ii}$, $b_{ij} = -\beta_{ij} / \beta_{ii}$ and $\tilde{x}_i$ is the relative abundance of species $i$ at equilibrium ($a_i$ is also known as the carrying capacity of the species). This equation allows us to estimate gLVM parameters (through $a_i$ and $b_{ij}$) from cross-sectional data, assuming that samples are at equilibrium, and absolute abundances are known ($x_i = m\tilde{x}_i$). To account for the fact that microbiome data provides relative abundances and biomass is typically not measured, BEEM-Static extends an EM framework to jointly estimate model parameters and biomass (**S1 Text**; [23]):

**Estimating biomass (E-step):** for a sample, the equation for each species $i$ provides an estimate for the biomass, and BEEM-Static takes the median of estimates across species as a robust estimator for the biomass of the sample:

$$m^{(T)} = \text{median}\left( -\frac{a_i^{(T-1)}}{\sum_{j=1}^{p} b_{ij}^{(T-1)} \tilde{x}_j} \right).$$

**Estimating model parameters (M-step):** BEEM-Static estimates model parameters ($a_i^{(T)}$ and $b_{ij}^{(T)}$) for each species $i$ with sparse regression (implemented with the 'glmnet' package in R) in iteration $T$:

$$\tilde{x}_i \sim \frac{1}{m^{(T)}} \cdot a_i^{(T)} + \sum_{j=1, j \neq i}^{p} b_{ij}^{(T)} \tilde{x}_j.$$

**Initialization and termination:** BEEM-Static initializes biomass values based on normalization factors from cumulative sum scaling (CSS [25]), with a user defined scaling constant as the median of biomass values (kept constant through EM iterations). The EM process is then run until the maximum number of iterations specified (200 by default) or until convergence when the median of relative changes in biomass values is $<10^{-3}$. Confidence values (Z-scores) for the final interaction matrix (non-zero off-diagonal entries) were calculated for each species

*i* using forward stepwise regression (implemented in R package "selectiveInference"; version 1.2.5; Akaike information criterion as the stopping criterion).

## 2.2 Statistical filters to detect violations of modeling assumptions

BEEM-Static uses the following filters to identify and remove samples that violate modeling assumptions and could thus impact model inference:

**Equilibrium filter**: to identify samples that may not be at equilibrium, BEEM-Static first predicts the abundances of species at equilibrium (based on the current model) for all species that are present:

$$\tilde{\boldsymbol{x}}^* = -\frac{1}{m^{(T-1)}}\left(\mathbf{B}^{-1}\boldsymbol{a}\right),$$

where $\boldsymbol{B}$ is the matrix form of $\hat{b}_{ij}^{(T)}$ (= estimate for $b_{ij}^{(T)}$), $\boldsymbol{a}$ is a column vector taking the value of $\hat{a}_i^{(T)}$ (= estimate for $a_i^{(T)}$) and $\tilde{\boldsymbol{x}}^*$ is a vector of predicted relative abundances at equilibrium for each species ($\tilde{x}_i^*$ is set to 0 if species $i$ is not present). Samples with median relative deviation above a user defined threshold ($\epsilon_1$) from these equilibrium values were then excluded as being potentially not at equilibrium (median($|\tilde{x}_i - \tilde{x}_i^*|/\tilde{x}_i$) > $\epsilon_1$, $\tilde{x}_i \neq 0$ and $\epsilon_1$ = 20% by default).

**Model filter**: to account for cases where some samples may come from an alternate gLVM, BEEM-Static calculates the median of squared errors for each sample $k$ with respect to the current model parameter estimates ($\hat{a}_i^{(T)}$ and $\hat{b}_{ij}^{(T)}$):

$$e_k = \underset{\tilde{x}_i \neq 0}{\text{median}}\left(\left(\tilde{x}_i - \left(\frac{1}{m^{(T-1)}}\cdot\hat{a}_i^{(T)} + \sum_{j=1, j\neq i}^{p}\hat{b}_{ij}^{(T)}\tilde{x}_j\right)\right)^2\right)$$

This is done in the M-step for each iteration and samples with large median squared error, i.e. $(e_k-\text{median}(e_k))/\text{IQR}(e_k)>\epsilon_2$ where IQR is the inter-quartile range and $\epsilon_2$ is a user defined parameter (default value of 3), are then removed for the next iteration's M-step.

## 2.3 Selecting shrinkage parameters for sparse regression

The shrinkage parameter λ in the sparse regression penalizes the number of parameters to avoid overfitting and is selected based on five-fold cross-validation in each iteration (selecting the value one standard error away from the best λ [26]). In the M-step of iteration $T$, a crude selection of $\lambda_c^{(T)}$ is made in BEEM-Static from a large range from $10^{-10}$ to $10^{-1}$, and then refined with a fine-grained sequence from $\lambda_c^{(T)}/10$ to $10\lambda_c^{(T)}$.

## 2.4 Generating simulated datasets

Simulated gLVM data was generated based on previously described procedures [16, 19]. Specifically, to parameterize distributions for generating model parameters, MDSINE [19] was used to estimate the mean and standard deviations of growth rates and inter-/intra-species interaction parameters from the *C. difficile* infection experiment data provided with the software. Growth rates and intra-species interactions were sampled from normal distributions (forced to be positive and negative respectively to model logistic growth). The interaction network structure was generated by randomly adding edges from one species to another (with probability ranging from 0.1 to 0.5) and the magnitude of the interactions was sampled from a normal distribution (with 0 mean and standard deviation estimated from real data as noted above). Initial abundances of $p$ (30) species were sampled from a uniform distribution (from 0.001 to the mean carrying capacity $\mu_i/\beta_{ii}$ of all the $p$ species), with each species having a

probability of π to be absent from a sample (π was estimated as the average rate of absence for the top *p* most prevalent species in all healthy gut microbiome profiles from the database curatedMetagenomicData [27]). A dataset with *n* samples was generated by numerically integrating the gLVM with the same parameters until equilibrium, starting with *n* different initial abundance profiles. The abundances of a random time point along the numerical integration (>20% away from the abundance at equilibrium for >50% of the species) was selected as a sample not at equilibrium. Poisson noise was added to the abundance of each species to simulate experimental variability.

## 2.5 Generating datasets based on growth curves and co-culture experiments

Microbiome profiles for co-culture experiments were taken from a previous study [20] and the relative abundances were scaled using the corresponding biomass measurements (OD600). Six species pairs were randomly selected, and one of the three conditions were randomly picked for each pair in each sample: (1) only the first species was present, (2) only the second species was present and (3) both species were present. For the first two cases, a random timepoint (last 6 timepoints near the equilibrium) was taken from the growth curve (measured by OD600) of species present. For the last case, the scaled abundances of the two species near the steady state (randomly taken from the three replicates) were used. Abundances were re-scaled to relative abundances and the process repeated to generate a dataset with 500 samples. The interaction matrix reported in Venturelli *et al* [20] was treated as the ground truth ("M-PW1-PW2").

## 2.6 Evaluation metrics

We computed the median of relative errors to assess the accuracy of predicted parameters as:

$$\text{median}\left(\frac{|\hat{\theta} - \theta|}{\max(|\hat{\theta}|, |\theta|)}\right),$$

where $\hat{\theta}$ and $\theta$ are the estimated and true parameters (***a***, ***b*** and ***m***) respectively. The area under the receiver operating characteristic curve (AUC-ROC) was computed for the interaction matrix. Z-scores were used to rank interactions (off-diagonal entries only) predicted by BEEM-Static. Sensitivity for predicting the signs of interaction was calculated as the fraction of interactions with correctly predicted signs in the true interaction matrix (non-zero off-diagonal entries only). Sign precision was computed as the fraction of interactions with correctly predicted signs.

## 2.7 Benchmarking with correlation-based methods

The following correlation-based methods (**Table 1**) for inferring interactions from microbiome data were tested.

The -log(p-value) or edge stability were used to rank CCREPE and SPIEC-EASI correlations, respectively, while the absolute values of correlation coefficients were used for the other methods. AUC-ROC was calculated from the lower triangle of the inferred correlation matrix (an inference was considered correct if there was an interaction between the corresponding species regardless of the interaction direction). Sensitivity and precision of signs were calculated as described above, excluding positive-negative interactions as they cannot be differentiated from positive-positive and negative-negative interactions using correlation analysis.

## 2.8 Analysis of gut microbiome data

Healthy gut microbiome profiles from the database curatedMetagenomicData were preprocessed and used as the standard dataset for learning gLVMs by removing (1) replicate samples,

**Table 1. Correlation methods tested.**

| Algorithm | Version | Parameters | Note |
|---|---|---|---|
| **Pearson and Spearman correlation** | | | Directly calculated from relative abundances |
| **CCREPE** [28] | v1.2.0 | 1000 iterations | Correction done for both Pearson and Spearman correlations |
| **SparCC** [29] | commit id: 9a1142c | Default | |
| **CCLasso** [30] | v1.0 | Default | |
| **REBACCA** [31] | https://tinyurl.com/ymdts4wy | nbootstrap = 50, B = 500, FWER = 0.01 | |
| **MInt** [32] | v1.0.1 | Default | |
| **SPIEC-EASI** [33] | v0.1.4 | lambda.min.ratio = 0.01, nlambda = 20, rep. num = 50 | Both "mb" and "glasso" algorithms |
| **BAnOCC** [34] | v1.0.1 | Default | |
| **gCoda** [35] | commit id: 584bd07 | Default | |

(2) timepoints other than the first timepoint in longitudinal studies, (3) samples under antibiotic treatment and (4) samples from infants. In addition, we included two validation datasets to evaluate different aspects of the model learned by BEEM-Static: (1) all samples from Raymond *et al* [36] to validate BEEM-Static's estimated growth and (2) samples from healthy infants (only the first timepoint for each subject) with ages below 12 months to evaluate BEEM-Static's biomass estimation. Both datasets were also used to validate BEEM-Static's ability to filter out samples violating model assumptions. To make the number of parameters tractable with the number of data points available, we only kept core species that were present (relative abundance >0.1%) in more than 30% of samples and subsequently removed samples where none of the core species were found, resulting in 42 core species (**S6 Fig**) and 4,617 samples overall. BEEM-Static was applied with the "model filter" ($\epsilon_2 = 0.9$) to learn two models (1,995 and 1,145 samples for each model) in two iterations, in which samples violating the filter were removed (1,477 samples removed in total). BEEM-Static was then rerun without the filter on samples assigned to each model separately to re-learn parameters.

### 2.9 Estimating *in situ* growth using BEEM-Static and GRiD

With BEEM-Static, *in situ* growth can be estimated as the deviation from equilibrium:

$$\hat{a}_i + \hat{m} \sum_{j=1}^{p} \hat{b}_{ij} \tilde{x}_j,$$

where $\hat{a}_i$, $\hat{m}$ and $\hat{b}_{ij}$ are estimated parameters. In addition, species replication rates for samples not under antibiotic treatment in Raymond *et al* [36] were estimated with the high-throughput mode of GRiD (v1.2.0; default parameters) [37]. Gut microbiome associated genomes provided with GRiD were used as references and read reassignment using pathoscope2 [38] was enabled (parameter "-p") to resolve ambiguous mappings.

## 3. Results

### 3.1 Accurate inference of ecological interactions from cross-sectional microbiome data

To evaluate if ecological interactions can be inferred from cross-sectional microbiome data (16S rRNA gene amplicon or whole metagenomic sequencing) we first began by conducting

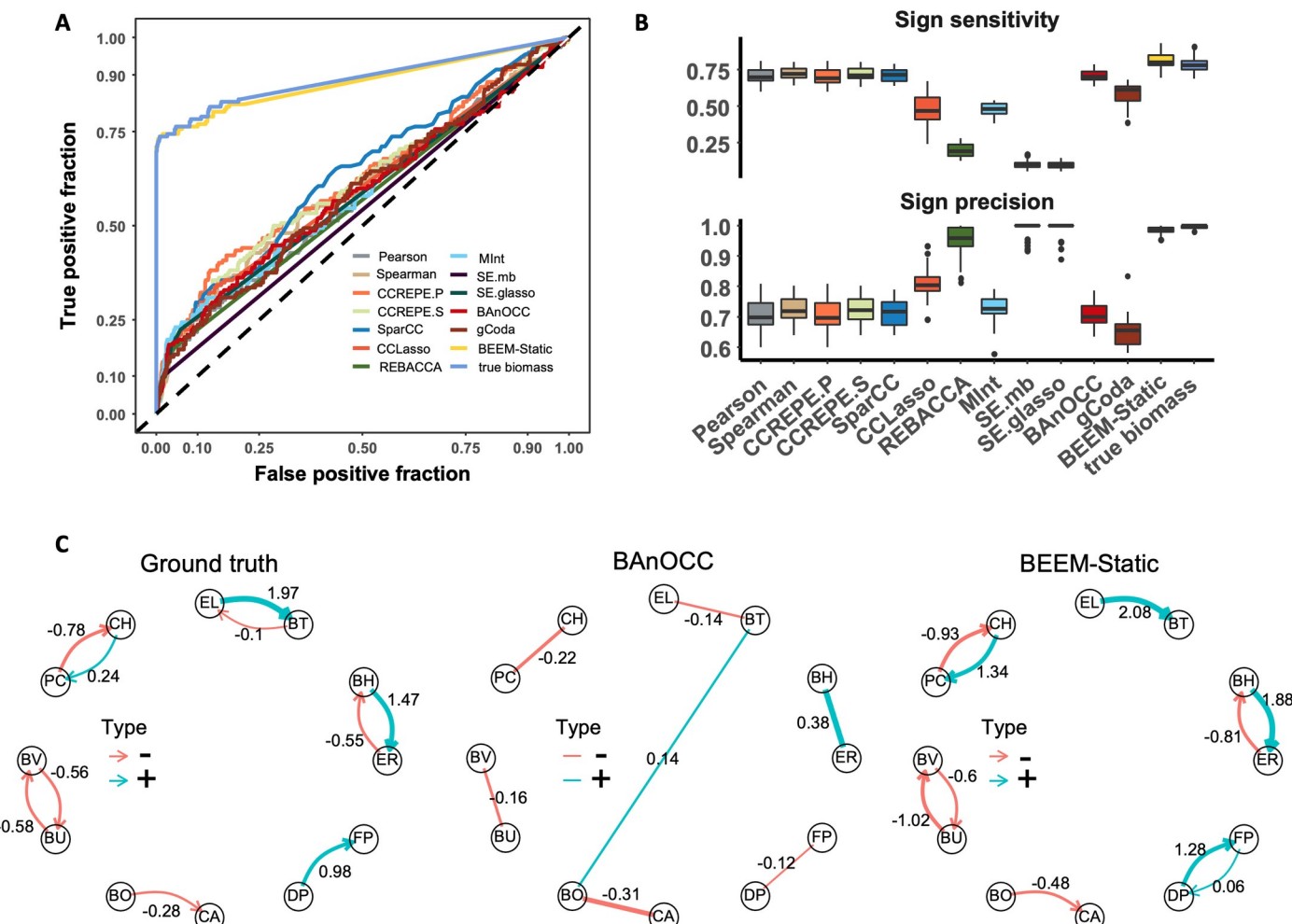

**Fig 2. Benchmarking performance for network structure and edge directionality.** Note that CCREPE has two versions with Pearson and Spearman correlations (CCREPE.P and CCREPE.S), while SPIEC-EASI using "mb" and "glasso" algorithms is represented as SE.mb and SE.glasso, respectively. (A) ROC curves for different correlation based methods, the regression method using true biomass values and BEEM-Static for one simulated dataset with 30 species and 500 samples. (B) Boxplots showing precision and recall for directionality/sign of interactions for 30 different simulated communities with 30 species and 500 samples each. (C) True interaction network for a synthetic community based on all-pair co-culture data (Ground truth), inferred correlation network by BAnOCC and inferred interaction network by BEEM-Static (numbers on edges represent gLVM parameter values or correlation coefficients). PC: *Prevotella copri*, BV: *Bacteroides vulgatus*, BU: *Bacteroides uniformis*, BO: *Bacteroides ovatus*, BT: *Bacteroides thetaiotaomicron*, FP: *Faecalibacterium prausnitzii*, BH: *Blautia hydrogenotrophica*, ER: *Eubacterium rectale*, CA: *Collinsella aerofaciens*, EL: *Eggerthella lenta*, DP: *Desulfovibrio piger*, CH: *Clostridium hiranonis*.

extensive benchmarks on complex simulated communities where the truth is known (using gLVMs, number of species = 30, equilibrium conditions; **Methods**). Consistent with prior reports [17, 18], we noted that all 10 state-of-the-art correlation-based methods provided performances that were only slightly better than random predictions in identifying true interactions, even while ignoring directionality (AUC-ROC<0.63; **Figs 2A** and **S1**). The ability to correct for compositionality of microbiome data (CCREPE, SparCC, CCLasso, REBACCA), or reduce false positives from transitive correlations (MInt, SPIEC-EASI, BAnOCC and gCoda), did not notably change performance compared to naïve correlation calculations (Pearson or Spearman) when inferring ecological interactions. In contrast, BEEM-Static was able to infer the interactions with notably higher sensitivity and specificity than all correlation-based methods (AUC-ROC = 0.88; **Figs 2A** and **S1**). Although BEEM-Static only used noisy relative

abundances for gLVM inference, its performance matched that of a positive control that assumes noise-free biomass values to scale relative abundances to absolute abundances (**Figs 2A** and **S1**, BEEM-Static vs. true biomass), an assumption that is unlikely to be met in realistic settings [23].

In addition to knowing that two species are interacting, a key aspect of ecological interactions is the directionality/sign of interactions, with correlation-based methods either exhibiting low sensitivity (REBACCA, SPIEC-EASI) or low precision (Pearson, Spearman, CCREPE, SparCC, CCLasso, MInt, BAnOCC, gCoda) despite the exclusion of predatory interactions (positive-negative) when calculating their performance (**Fig 2B**; **Methods**). BEEM-Static addresses this issue by providing high sensitivity (>80%) and precision (nearly 100%) using cross-sectional microbiome data (**Fig 2B**). These observations were recapitulated in a wide range of simulated datasets with varying network structure and edge sparsity, highlighting BEEM-Static's robustness (**S2** and **S3 Figs**). Furthermore, BEEM-Static provides estimates for biomass values that were found to be consistently accurate (relative error <10%) and can be used to provide meaningful biological insights [23].

We extended the evaluations to experimental data, using all-pair co-culture and isolate growth curves for 12 species [20] to create synthetic communities where the interaction network is known, albeit sparse (**Fig 2C**; Ground truth). Not surprisingly, recapitulating the structure of such a simple interaction network was not difficult for most correlation-based methods, with BAnOCC having the best performance overall (AUC-ROC = 0.9, **S4 Fig**). However, determining directionality of interactions was still a challenge, despite the simplicity of the network. For example, in the case of BAnOCC the commensalistic interaction between DP and FP was captured as a negative correlation, while the predatory interaction between BH and ER was captured as a positive correlation (**Fig 2C**; BAnOCC). BEEM-Static, on the other hand, was able to capture all interactions and their directionality correctly, except for one false positive (FP to DP) and one false negative (BT to EL) involving interactions with weak strength (**Fig 2C**; BEEM-Static). BEEM-Static's utility in such datasets was consistently observed in comparison to correlation-based methods with AUC-ROC close to 1 (**S4 Fig**).

## 3.2 Statistical filters in BEEM-Static provide robustness to violations in modeling assumptions

While the simulations in the previous section account for experimental errors, they assume that all samples come from equilibrium states, an unlikely situation for most real datasets. Relaxing this assumption, we noted that with as little as 5% non-equilibrium samples, AUC-ROC decreased by >10%, and with 15% non-equilibrium samples AUC-ROC performance degraded to match that of correlation-based methods (Naïve algorithm, **Figs 3A** and **S5A**; **Methods**). Incorporating a statistical filter in BEEM-Static that compares estimated species relative abundances at equilibrium with observed abundances (equilibrium filter; **Methods**) helped identify samples that were not at equilibrium with high sensitivity and specificity (**S5A Fig**). This in turn allowed BEEM-Static to be robust to having nearly half of the samples (45%) in the dataset being at a non-equilibrium state (performance reduction <5%; **Figs 3A** and **S5A**, BEEM-Static).

We next investigated the impact of relaxing the "universal model" assumption i.e. that all samples have the same ecological conditions and model parameters. This assumption may not hold true in many real-world settings (e.g. gut microbiome samples from different enterotypes [24]), and as expected relaxing it had a strong impact (30% reduction in AUC-ROC) even with slight deviations (5%, Naïve algorithm; **Figs 3B** and **S5B**; **Methods**). In addition, AUC-ROC performance continued to decrease beyond 60% even after nearly half the samples (40–45%)

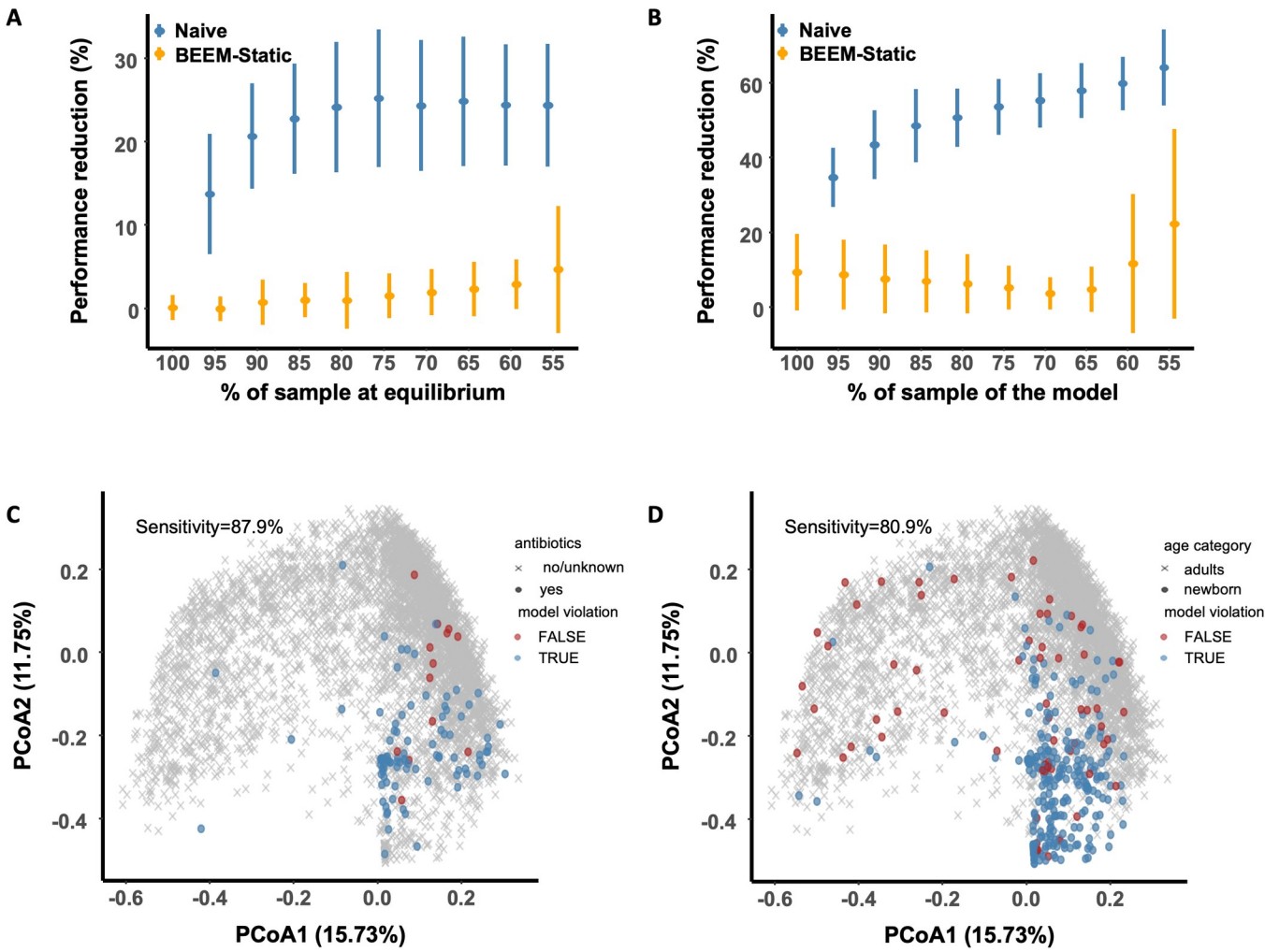

**Fig 3. BEEM-Static robustly filters samples violating modeling assumption in simulated and real datasets.** (A-B) Performance reduction for BEEM-Static (with filters) and the naïve algorithm (without filtering of samples) as the percentage of samples at (A) equilibrium or (B) generated from the main model, decreases. Reduction is measured relative to BEEM-Static with no filters and all data from the model and at equilibrium. Points denote the means while error bars denote the standard deviation across 30 simulations each. (C-D) Principal coordinates plots (Bray-Curtis dissimilarity) representing gut microbiome taxonomic profiles from 4,617 samples. Points represent samples from individuals taking antibiotics (C) or from newborn infants (D), while crosses represent samples from adults who are not undergoing antibiotic treatment. Points that were filtered by BEEM-Static are colored blue and red otherwise.

were derived from a different model. To address this, BEEM-Static implements a filter that identifies samples that have poor goodness-of-fit to the current model (model filter; **Methods**) and excludes them in subsequent iterations of model inference. This approach was found to provide robustness to up to ~40% of samples from a different model (performance reduction <20%, BEEM-Static; **Figs 3B** and **S5B**).

Finally, we employed real microbiome datasets to test BEEM-Static's robustness where a subset of samples is known to violate modeling assumptions i.e. some subjects undergoing oral antibiotic treatment or samples from newborn infants, where the majority of samples are from adults who are not undergoing antibiotic treatment. BEEM-Static was able to identify such samples with high sensitivity (>80%) using its model filter (**Fig 3C** and **3D**), and the filtered samples were significantly enriched for those from antibiotic-treated adults and infants (Fisher's Exact test p-value<$10^{-22}$).

### 3.3 BEEM-Static analysis of human gut microbiomes identifies distinct ecological configurations

To further assess BEEM-Static's utility we evaluated the concordance of parameters learnt during the training process with orthogonal information for a large human gut microbiome dataset (N = 4,617; **Methods**). In particular, we noted that biomass estimates from BEEM-Static were significantly higher for adults versus newborn infants (~2×; Wilcoxon test p-value $<10^{-15}$; **Fig 4A**), consistent with our understanding of a maturing gut microbiome [39, 40]. Additionally, we used deviations from equilibrium ($dx_i(t)/dt = 0$) to estimate instantaneous growth (population increase or decrease) of each species in each sample (**Methods**), and assessed concordance with an *in silico* approach to estimate DNA replication rates [37]. Despite the fact that growth rates are also impacted by death rates, we observed that species predicted to grow based on BEEM-Static analysis were also found to have significantly higher DNA replication rates (GRiD values; Wilcoxon test p-value = $3\times10^{-4}$; **Fig 4B**).

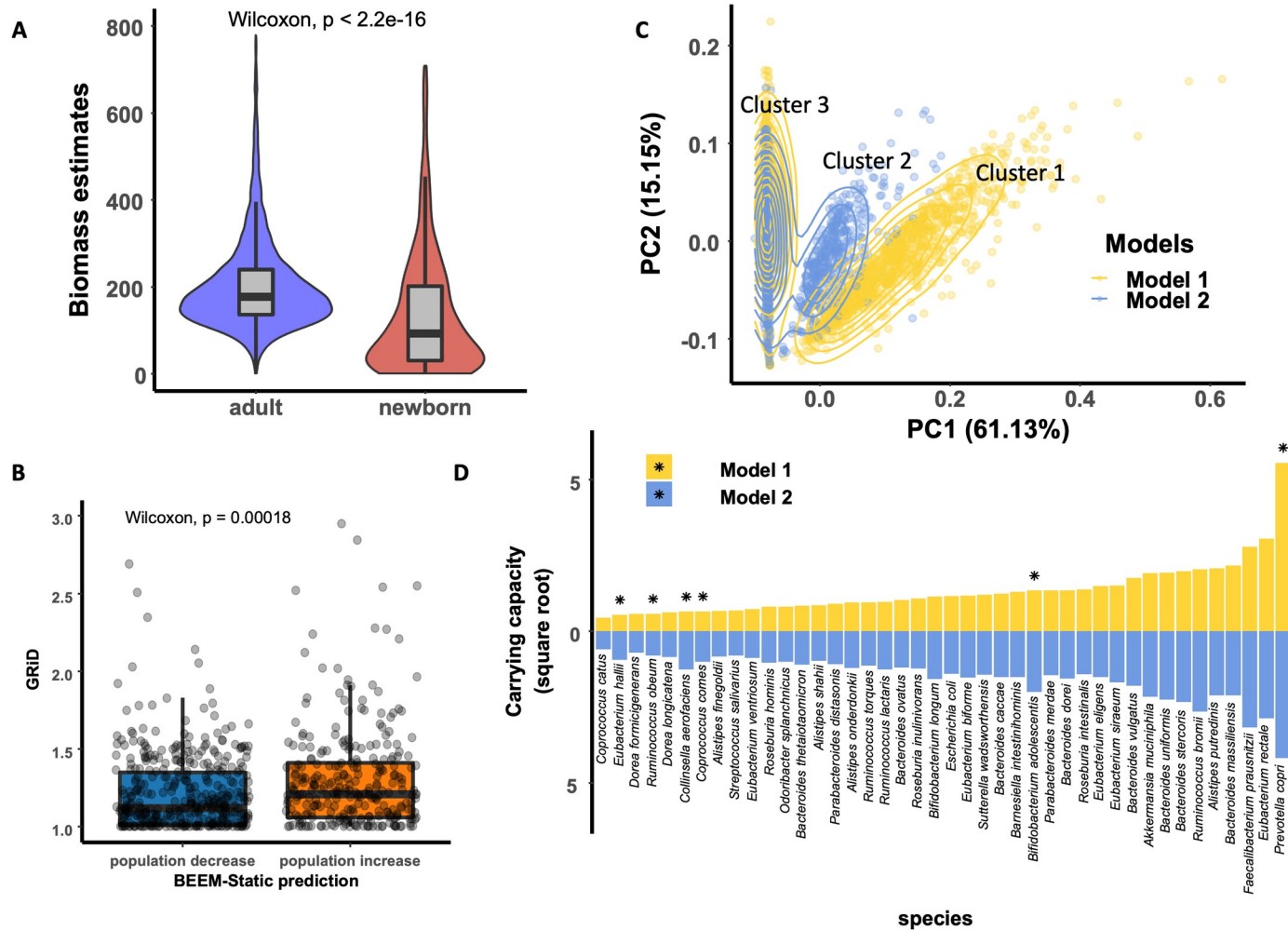

**Fig 4. Analysis of human gut microbiomes with BEEM-Static.** (A) Violin plots showing the significant difference in BEEM-Static biomass estimates for adults and newborn infants. (B) Boxplots showing differences in DNA replication rates (GRiD estimates) for species predicted to decrease and increase in population size, respectively, by BEEM-Static. Each point represents one species in a sample. (C) Scatter plot showing the first two components from a principal component analysis of equilibrium abundances for samples (as predicted by BEEM-Static). (D) Predicted carrying capacities (square root transformed) of species in the two models. Species with divergent carrying capacities (ratio is >2 standard deviations from 1) in the two models are marked with stars.

Using an iterative approach to train a new model on samples excluded from the first model, we observed that BEEM-Static classifies a majority of the samples to two models (1995 for model 1 and 1145 for model 2; **S1 Table**). Visualizing the expected microbial composition for each sample at equilibrium, we noted three distinct clusters in a principal component analysis plot (**Fig 4C**), where samples in clusters 1 and 2 largely correspond to profiles from models 1 and 2, respectively. Analysis of carrying capacities highlighted that the clusters were defined by *Prevotella copri*, which has higher carrying capacity in cluster 1 vs 2 and is absent in samples from cluster 3 (**Fig 4D**). Several other species were also found to have divergent carrying capacities in the two models, including *Bifidobacterium adolescentis*, *Collinsella aerofaciens* and *Coprococcus comes* (enriched in model 2), some of which have been shown to be associated with fiber intake [41, 42]. Overall the ecological models identified here have distinct sets of interactions (**S1 Table**) and do not appear to match earlier definitions of enterotypes [24] based on principal coordinate analysis and clustering (S**7 Fig**).

## 4. Discussion

As microbiome research increasingly moves from descriptive studies to those that seek to provide a mechanistic understanding of microbial communities, the ability to infer microbial interactions from microbiome data is an important capability. In particular, the directionality and sign of interactions provide biologically interpretable information that is missed by correlation-based approaches. BEEM-Static provides an alternative avenue to infer this, with the caveats that it assumes a specific model (generalized Lotka-Volterra) for community dynamics and removes rare and low abundance species with limited number of samples in practice. In addition, as we show here, other values obtained from BEEM-Static models can have utility, including the strength of interactions, biomass estimates, deviation from equilibrium, and fit to model.

In addition to accounting for relative abundance estimates from microbiome data, the statistical filters employed by BEEM-Static make it robust to some of the violations in model assumptions that can be expected in real datasets. These features make BEEM-Static widely applicable, and also extends the use of ecological models with microbiome data. For instance, our analysis of large public microbiome datasets provides an alternate perspective to the discussion on microbial enterotypes [24] and universality of microbiome dynamics [43]. The ecological types observed here are characterized by distinct carrying capacities that might be a function of the environment (e.g. host factors or diet). Fiber rich diets are known to have a strong impact on the gut microbiome [42] and have been linked to some of the species with differential carrying capacities in our models [41]. We anticipate that the incorporation of such environmental factors into future models would be an exciting avenue to study their influence on microbial community structure *in vivo*. Finally, hybrid methods that learn models from both longitudinal and cross-sectional data represent another promising direction to explore for studying general and individual specific microbiome dynamics [23].

## Supporting information

**S1 Fig. Performance comparison for determining interaction network structure.** Boxplots show AUC-ROC values for 30 different simulated communities with 30 species and 500 samples each.
(TIFF)

**S2 Fig. Robustness of BEEM-Static performance across different network structures.** Boxplots showing relative error for (A) parameter estimates for biomass (scaled to have the same

median as the ground truth) and carrying capacity, and (B) interaction network AUC-ROC as well as sign recall and precision, based on 30 simulations. Diamonds mark median performance using true biomass values. Different network structures (except "random") were generated with the SPIEC-EASI R package.
(TIFF)

**S3 Fig. Robustness of BEEM-Static performance with varying edge densities.** Boxplots showing relative error for (A) parameter estimates for biomass (scaled to have the same median as the ground truth) and carrying capacity, and (B) interaction network AUC-ROC as well as sign recall and precision, based on 30 simulations. Diamonds mark median performance using true biomass values.
(TIFF)

**S4 Fig. Benchmarking on synthetic communities created based on co-growth experimental data.** Boxplots show the performance of various methods based on 30 replicates that use randomly selected interacting pairs.
(TIFF)

**S5 Fig. Utility of BEEM-Static filters for avoiding performance reduction in the presence of model violations.** Results are shown for increasing proportion of samples that are, (A) not at equilibrium or (B) not from the main model. The naïve algorithm is without filtering of samples and performance reduction is measured relative to BEEM-Static with no model violations for the data. Points show means while error bars show standard deviation across 30 simulations.
(TIFF)

**S6 Fig. Percentage of the gut microbiome represented by the 42 core species used by BEEM-Static for modeling with the curatedMetagenomicData dataset.**
(TIFF)

**S7 Fig. Ecological models from BEEM-static represent distinct configurations from enterotypes.** Principal coordinates plot (Bray-Curtis dissimilarity) based on gut microbiome taxonomic profiles. The contour lines highlight the two different regions where points aggregate. Points are colored by the BEEM-static models that they belong to (unassigned samples not shown), showing that while there appears to be some enrichment, model ids and enterotypes do not show a 1–1 correspondence.
(TIFF)

**S1 Table. Parameters estimated by BEEM-Static for the two models learned from the curatedMetagenomicData dataset.**
(XLSX)

**S1 Text. Derivation of the expectation-maximization algorithm and training convergence.**
(PDF)

## Acknowledgments

We thank Dr. Swaine Chen, Dr. Rohan Williams and Dr. Jonathan Teo for their constructive comments and feedback on this work.

## Author Contributions

**Conceptualization:** Chenhao Li, Kern Rei Chng, Niranjan Nagarajan.

**Data curation:** Tamar V. Av-Shalom, Jun Wei Gerald Tan, Junmei Samantha Kwah.

**Formal analysis:** Chenhao Li, Tamar V. Av-Shalom, Jun Wei Gerald Tan, Niranjan Nagarajan.

**Funding acquisition:** Niranjan Nagarajan.

**Investigation:** Chenhao Li, Kern Rei Chng, Niranjan Nagarajan.

**Methodology:** Chenhao Li, Niranjan Nagarajan.

**Project administration:** Kern Rei Chng.

**Resources:** Niranjan Nagarajan.

**Software:** Chenhao Li, Tamar V. Av-Shalom, Jun Wei Gerald Tan.

**Supervision:** Niranjan Nagarajan.

**Visualization:** Chenhao Li, Tamar V. Av-Shalom.

**Writing – original draft:** Chenhao Li, Niranjan Nagarajan.

**Writing – review & editing:** Chenhao Li, Niranjan Nagarajan.

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
