## [Decision Letter · Decision Letter 0]

10 May 2021

Dear Dr. Nagarajan,

Thank you very much for submitting your manuscript "BEEM-Static: Accurate inference of ecological interactions from cross-sectional metagenomic data" for consideration at PLOS Computational Biology.

As with all papers reviewed by the journal, your manuscript was reviewed by members of the editorial board and by several independent reviewers. In light of the reviews (below this email), we would like to invite the resubmission of a significantly-revised version that takes into account the reviewers' comments.

We cannot make any decision about publication until we have seen the revised manuscript and your response to the reviewers' comments. Your revised manuscript is also likely to be sent to reviewers for further evaluation.

Sincerely,

Nicola Segata

Associate Editor

PLOS Computational Biology

Jason Papin

Editor-in-Chief

PLOS Computational Biology

Reviewer's Responses to Questions

**Comments to the Authors:**

Reviewer #1: Li et al. developed a tool called "BEEM-Static" to infer microbial interactions from cross-sectional metagenomic data. These interactions are inferred using a generalized Lotka-Volterra model (gLVM). The authors highlight that in contrast to correlation-based methods BEEM-Static is able to infer directionality in addition to presence of the interactions. Further, they provide estimates of the sample biomass. They benchmarked their algorithm against 10 correlation-based methods using simulated datasets, simple pairwise experimental co-culture data and a large cross-sectional metagenomic dataset. BEEM-Static was robust to violations of model assumptions (equilibrium states and model differences) in these tests. While their method is interesting and the benchmarking results show improvements over other tools, I have several concerns:

Major concerns

• What is the main difference to the original BEEM algorithm? The filter functions? Please add more details on that to the introduction.

• Further details about model training should be provided. For example, additional figures showing how well the model converged by the end of training.

• The parameter inference method used in the paper is a coordinate descent algorithm, a general form of EM-algorithm. For the EM algorithm, it is necessary to write the log-likelihood function and formulate the E-step and M-step in this framework. That way, the M-step is maximizing the lower bound of the log-likelihood function which is an essential guarantee for the model to converge. In the general coordinate descent, it is necessary to prove that the function is bounded in order to show that the algorithm will converge. This is missing currently.

• In the large-scale metagenomic dataset, only 42 species passed the filtering. This number is very small for human gut microbial communities and a large fraction of the microbial community is not considered in many samples. I would suggest the authors provide a barplot showing the fraction of considered microbes in each sample.

• The authors didn't report how many samples are found invalid in the large-scale metagenomic dataset. They showed that the model still works well when up to 40% of samples are at non-equilibrium. Without the number of how many samples are predicted as "invalid" in the metagenomic dataset, it is uncertain if this criteria is met for the large metagenomic dataset.

• I would recommend a graphical overview of BEEM-static as one of the first figures, including input, output and potential conclusions that can be inferred from the results.

• I’m curious what the rationale is behind the biomass estimation. Why should it be possible to infer biomass based on relative abundances?

• Last paragraph of results section: Why do you point out fibre might play a role here? Without the context from the cohort, this seems a bit random. Please describe the real world datasets you used briefly (Healty cohort? Disease?) and why fibre is relevant.

• Why did you expect that enterotypes play a role in this or are affected by interactions?

Minor concerns:

• I don’t see why their approach is only applicable to metagenomic data. The abundance profiles from 16S data should work equally well.

• It would be interesting to verify the model's prediction on the equilibrium state of the metagenomic samples from the other time points. Does the next timepoint deviate from the current one? Does the second time point of the samples predicted as invalid deviate more than those predicted as valid? That would provide further confidence for the model's performance.

• Given a model, there will be multiple equilibrium states (the eigenvectors of the state matrix). It would be interesting to show how these equilibrium states are distributed in the large cohort.

• Some terms should be defined in the introduction, such as equilibrium. Examples for bacterial interactions of the different types would be helpful too (pos-pos, neg-neg, pos-neg).

• p. 15 (on my version): Define the abbreviation for the species involved in the interaction (DP, FP, ...).

• Fig 1C: Explain in the legend what those numbers in connection with the interactions mean.

• Fig 3C: The axis relative to the origin would make more sense to me. Also, I’m curious why the first PC explains so much variance? How were the clusters computed?

Reviewer #2: This manuscript introduces a new bioinformatics algorithm,

"BEEM-Static", to infer ecological interactions from cross-sectional

metagenomic data. The work is motivated by the current needs in

obtaining deeper ecological understanding of the systems underlying

observation in microbiome research in general, and in microbial

metagenomics in particular. Currently popular methods for

computational inference of microbial interactions are based on

observed co-abundance patterns but often lacking ecological

insight. This paper aims to demonstrate that ecological structure can

be recovered from cross-sectional metagenomic data. The method is an

EM algorithm (BEEM-Static) that utilizes a standard ecological (gLV)

interaction model. Application requires sample filtering but this is

justified based on the modeling assumptions: in particular, the

authors show that gLV parameters can estimated directly from

cross-sectional data, when it is being assumed that 1) the system is

close to equilibrium, and 2) absolute abundances are known. This has

considerable practical value because it helps to obtain ecologically

relevant insights from dynamical systems in a setting where the lack

of long and dense time series has so far limited the

investigations. Comparison with relevant alternatives demonstrates

remarkable improvements in the ability to detect interaction and its

directionality reliably (based on AUC/ROC scores). It is also shown

that the method is relatively robust to outliers and deviations from

stationarity or model assumptions, so that the first assumption seems

feasible. The second assumption is not generally available but the

authors propose a way to estimate this computationally, and advances

in measurement techniques may improve this further over time; the main

contribution in this work is the method itself. An interesting

demonstration of the model performance is provided by analysing

community stability in a well-known public human gut microbiome data

set, and the proposed model is potentially applicable to studying key

aspects of community dynamics across a vast spectrum of microbiome

data sets, and the results can be experimentally testable in many

cases if suitable experimental time series can be collected.

The manuscript is technically sound and written in a fluent and easily

understandable English. Experiments and statistical analyses have been

conducted rigorously and the conclusions are supported by the

data. Relevant literature is cited, and the relevant recent methods

have been included in the comparisons.

All data and code are fully available but I have not tried to

replicated the analysis. The associated R package seems good quality

and implemented according to prevailing standards for R package

development.

I only have the following minor notes:

* Minor

- The ecological models do not correspond to previously reported

enterotypes (in particular Fig S6). However, enterotypes - as

visualized on 2D PCoA space - are very broad community-level

clusters. It is possible that there are more refined subcommunities

that would be better visible on further PCoA dimensions or in a

focused subcommunity analysis and these might correspond better with

the previously reported enterotypes that are said to be driven by

specific bacterial groups. It would be informative to include some

more discussion on the possible connections. In particular,

Prevotella has been shown to have bimodal/bistable abundance

distribution at the population level, and if the BEEM-Static model

does not match this finding, this would be somewhat unexpected and

potentially indicate problems in the BEEM-Static model.

- The method is promoted as a new method for metagenomic data; but it

seems to me that this applies more generally to ecological community

profiling data, including 16S and potentially not even limited to

microbial ecological communities. Some discussion on the wider scope

might be warranted.

- The model is applied for core species that are abundant and

prevalent; applicability to lower abundance and rare groups is a

possible limitation and could be more clearly stated in Discussion.

- It would be informative and helpful to show how the term in M-step

inside the median (a / \\Sigma bx) can be derived for estimating

absolute abundances.

- The "EM framework" is not exactly the classical EM algorithm

although very close conceptually; reconsider naming e.g. "iterative

EM-type framework" or otherwise acknowledging the difference.

- The bullet point list in 2.7 might be better written out as text or

incorporated as a table?

- principle component analysis (common misspelling) -> should be:

principal component analysis; same for principal coordinate analysis

**Have the authors made all data and (if applicable) computational code underlying the findings in their manuscript fully available?**

Reviewer #1: Yes

Reviewer #2: Yes

PLOS authors have the option to publish the peer review history of their article (what does this mean?). If published, this will include your full peer review and any attached files.

Reviewer #1: No

Reviewer #2: **Yes: **Leo Lahti
---

## [Decision Letter · Decision Letter 1]

11 Aug 2021

Dear Dr. Nagarajan,

We are pleased to inform you that your manuscript 'BEEM-Static: Accurate inference of ecological interactions from cross-sectional microbiome data' has been provisionally accepted for publication in PLOS Computational Biology. There are a couple of minor points still highlighted by one of the reviewer, but I trust you will address them in the final version of the manuscript.

Best regards,

Nicola Segata

Associate Editor

PLOS Computational Biology

Jason Papin

Editor-in-Chief

PLOS Computational Biology

Reviewer's Responses to Questions

**Comments to the Authors:**

Reviewer #1: The authors have addressed my concerns and I have no further comments.

Reviewer #2: The manuscript has improved, and the responses are adequate.

The remaining comments:

- In review responses the authors cite the Remien et al. preprint (https://www.biorxiv.org/content/10.1101/463372v1.full) discussing the estimation of absolute abundances; in fact Remien et al. show the expected result that absolute abundances cannot be fully identified from compositional data. This is in contrast with the approach in BEEM-Static, which provides an estimate of absolute abundances based on compositional data. The authors have mentioned, however, that this is only an approximation. It is important to make sure that the inability to fully estimate absolute abundances from compositional data are mentioned clearly when discussing the limitations of the method. Better estimates may become available as absolute quantification methods are developing, further increasing the potential value of this method. therefore I do not consider the inaccessibility of absolute estimates a major limitation in this case.

- Text is understandable but I suggest to pay attention to final proof-reading and figure quality

**Have the authors made all data and (if applicable) computational code underlying the findings in their manuscript fully available?**

Reviewer #1: Yes

Reviewer #2: Yes

PLOS authors have the option to publish the peer review history of their article (what does this mean?). If published, this will include your full peer review and any attached files.

Reviewer #1: No

Reviewer #2: **Yes: **Leo Lahti

---

## [Editor Report · Acceptance letter]

1 Sep 2021

PCOMPBIOL-D-21-00585R1 

BEEM-Static: Accurate inference of ecological interactions from cross-sectional microbiome data

Dear Dr Nagarajan,

I am pleased to inform you that your manuscript has been formally accepted for publication in PLOS Computational Biology. Your manuscript is now with our production department and you will be notified of the publication date in due course.

With kind regards,

Andrea Szabo
